# Anatomical Study of Sites and Surface Area of the Attachment Region of Tibial Posterior Tendon Attachment

**DOI:** 10.3390/ijerph192416510

**Published:** 2022-12-08

**Authors:** Inori Uchiyama, Mutsuaki Edama, Hirotake Yokota, Ryo Hirabayashi, Chie Sekine, Sae Maruyama, Mayuu Shagawa, Ryoya Togashi, Yuki Yamada, Ikuo Kageyama

**Affiliations:** 1Institute for Human Movement and Medical Sciences, Niigata University of Health and Welfare, Niigata 950-3198, Japan; 2Department of Anatomy, School of Life Dentistry at Niigata, Nippon Dental University, Niigata 951-8151, Japan

**Keywords:** tibialis posterior tendon dysfunction, attachment site, surface area region

## Abstract

Background: The purpose of this study was not only to examine the attachment site but also to quantify the effect of the tibialis posterior tendon (TPT) on each attachment site by examining the surface area of the attachment region. Methods: We examined 100 feet from 50 Japanese cadavers. The TPT attachment to the navicular bone (NB), medial cuneiform bone (MCB), and lateral cuneiform bone (LCB) were set as the main attachment sites (Type I). The attachment seen in Type I with the addition of one additional site of attachment was defined as Type II. Furthermore, surface area was measured using a three-dimensional scanner. Results: Attachment to the NB, MCB, and LCB was present in all specimens. The TPT attachment to the NB, MCB, and LCB comprised 75.1% of total attachment surface area. The ratio of the NB, MCB, and LCB in each type was about 90% in Types II and III, and 70–80% in Types IV–VII. Conclusion: The quantitative results demonstrated the NB, MCB, and LCB to be the main sites of TPT attachment, although individual differences in attachment sites exist, further developing the findings of previous studies.

## 1. Introduction

The tibialis posterior (TP) plays a very important role during gait as the primary dynamic stabilizer of the medial longitudinal arch [1]. A typical disorder of the tibialis posterior tendon (TPT) is TPT dysfunction (TPTD), which occurs when the TPT becomes inflamed or torn [2]. Previously, although TPTD was thought to be secondary to overuse resulting in acute and chronic tendinitis, more recent histopathological evidence has revealed this pathology as a degenerative tendinosis with a nonspecific reparative response to tissue injury [3]. TPTD is the prevailing cause of adult acquired flatfoot deformity (AAFD), and it is characterized by collapse of the medial longitudinal arch [4]. An anatomical understanding of the TPT is therefore important not only for understanding the supporting mechanism of the medial longitudinal arch but also for the evaluation and treatment of TPTD. Therefore, many anatomical studies of TPT have been reported [4,5,6,7].

Regarding the sites of TPT attachment, Musial [6] described a permanent, main TPT insertion onto the navicular bone (NB), medial cuneiform bone, intermediate cuneiform bone (ICB), lateral cuneiform bone (LCB), and 2nd–3rd metatarsal bones (2–3 MB). However, among the 122 feet examined, different sites of attachment were found for the accessory bundles, including the cuboid bone (116 feet), 4th metatarsal bone (4 MB) (106 feet), flexor hallucis brevis (104 feet), calcaneus and CB (36 feet), peroneus longus muscle (20 feet), and 5th metatarsal bone (5 MB) (four feet). In addition, Bloome et al. [5] classified TPT into three fibers, anterior, middle, and posterior bundles in an investigation of 11 fresh frozen cadaver feet and examined the sites of attachment for each bundle. The anterior bund was reported to attach to the navicular tuberosity, inferior capsule of the naviculocuneiform joint, and the inferior surface of the MCB. The middle bund attached to the ICB, LCB, CB, 2nd–5th metatarsal bones, flexor hallucis brevis, and peroneus longus tendon. The posterior bund attached to the sustentaculum tali and spring ligament. Furthermore, recently, Olewnik [7] investigated the site of TPT attachment in 80 fixed cadaver feet, defining attachment to the NB and MCB as Type I, and categorizing Types II–IV and subtypes A–C according to additional sites of attachment. Willegger et al. [4] showed that the major site of TPT attachment in 41 fixed cadaveric feet was the NB. Branching and attachment to up to eight bones was reported among any of the MCB, ICB, LCB, CB, calcaneus, and 1st–5th metatarsal bones. In addition, Park et al. [8] reported four types of TPT insertions classified using 118 feet, wherein the most common type was type 4 (quadruple insertions, 78 feet, 66.1%), which was divided into four new subtypes that were not defined in the previous classification. The second most common type was type 3 (triple insertions, 25 feet, 21.2%) with three subtypes, including the new subtype. Type 2 was found in 13 feet (11%), and the rarest type was type 1 (2 feet, 1.7%), wherein the main tendon was only attached to the navicular bone and the medial cuneiform bone.

As these previous studies show, no consensus has been reached regarding sites of TPT attachment [4,5,6,7,8]. Possible obstacles to the development of such a consensus include ethnic differences, differences in laterality, and differences in numbers of samples [4]. In addition, while previous studies [4,5,6,7,8] have considered that the NB, MCB, and LCB are the main sites of TPT attachment, the degree of contribution of these attachment sites has not been quantified, and whether they contribute significantly to the actions of the TPT remains unknown. Main attachment sites thus need to be clarified by calculating and comparing the surface area ratio of individual attachment sites.

The purpose of this study was not only to examine the attachment site but also to quantify the effect of TPT on each attachment site by examining the surface area of the attachment region.

## 2. Materials and Methods

### 2.1. Cadavers

We examined 100 feet from 50 Japanese cadavers (mean age at death, 80 ± 11 years; 56 sides from 28 men, 44 sides from 22 women; 50 right sides, 50 left sides) that had been switched to alcohol after placement in 10% formalin. No legs showed any sign of previous major surgery around the ankle. This study was approved by the ethics committee at our institution (approval number: 18867).

### 2.2. Measurement Procedures

The procedure for dissecting the TPT is described below. Isolated specimens of the leg were created by transection about 10 cm above the ankle. The skin and subcutaneous tissue were removed and the peroneus longus and the peroneus brevis muscles and TPT were carefully dissected out and inspected. The plantar aponeurosis was carefully dissected from the flexor digitorum brevis and removed. The first, second, and third plantar layers were then dissected and excised. Upon reaching the fourth layer, to clarify sites of attachment to the TPT, the flexor halluces brevis, flexor digiti minimi brevis, adductor hallucis, opponens digiti minimi, first–fourth dorsal interossei, first–third plantar interossei, part of peroneus longus, the lateral Lisfranc ligament, and the metatarsal plantar ligament were dissected and excised (Figure 1A). Based on a previous study [7], the regions of attachment to the NB, MCB, and LCB were set collectively as the main attachment site (Type I). The attachment seen in Type I with the addition of one additional site of attachment was defined as Type II, which thus showed attachment at a total of four sites. In the same manner, classification up to Type VII was applied according to the number of additional attachment sites. Furthermore, the attachment site area was identified by peeling away any adherent tissue, then coloring the attachment site with a pencil (Figure 1B). The surface area was then measured using a three-dimensional (3D) scanner (EinScan Pro HD; SHINING 3D, Hangzhou, China) (manufacturer’s specifications, measurement precision of 0.04 mm) to produce a 3D foot sample (Figure 1C). The resulting data were read into Geomagic Freeform 2021 design software (3D SYSTEMS), and the boundary of the attachment site was drawn as a curve with a pen-type device (Touch; 3D SYSTEMS) (Figure 1D). Surface area was then calculated using Rhinoceros7 3D software (McNeel) (Figure 1E). Each attachment site was measured twice to allow calculation of the mean value and standard deviation. The ratio of each adhered area was calculated so that the sum of adhered areas for each sample was 100%. All measurements were carried out by the same physical therapist (I.U.).

The reliability of surface area measurement by 3D scanner was calculated using the intraclass correlation coefficient (ICC) (1,2) for 10 of 100 feet, yielding an ICC of 0.993. In this study, measurement of surface area showed almost perfect reliability, consistent with previous results [9].

### 2.3. Statistical Analysis

Statistical analyses were performed using SPSS version 24.0 (SPSS Japan, Tokyo, Japan). Pearson’s chi-squared test was used to compare differences in insertion location between sex and laterality. The level of significance was set at 5%.

## 3. Results

### 3.1. Sites and Ratios of TPT Attachment 

Attachment to the NB, MCB, and LCB was present in all specimens (100%). Attachment to the ICB was seen in 25 feet (25%), and to the CB in 66 feet (66%). Among 1–5 MBs, 4 MB was the most commonly attached, at 95 feet (95%) (Table 1).

### 3.2. Type Classification for Site of TPT Attachment

With NB, MCB, and LCB considered together as the most basic attachment site (Type I), seven types were classified according to the number of additional attachment sites, with subtypes A–D defined according to the types of additional attachment sites (Figure 2, Table 2). Of the seven types, Type IV was the most common, seen in 29 feet (29%), while Type VII was the least common, in 2 feet (2%). No cases of Type I were identified (Table 2). In addition, Types II and III could be classified into three subtypes (A–C), Types IV and V could be classified into four subtypes (A–D), and Type VI could be classified into two subtypes (A and B).

### 3.3. Differences in Sex and Laterality by Type (Table 3)

There were no significant differences between male and female (*p* = 0.613) and between the right and left legs (*p* = 0.582) for each Type.
ijerph-19-16510-t003_Table 3Table 3Differences in laterality and sex for each type.TypeMaleFemaleRightLeftTotalI0 (0)0 (0)0 (0)0 (0)0 (0)II6 (10.7)5 (11.4)6 (12.0)5 (10.0)11 (11.0)III15 (26.8)11 (25.0)13 (26.0)13 (26.0)26 (26.0)IV17 (30.4)12 (27.3)12 (24.0)17 (34.0)29 (29.0)V13 (23.2)10 (22.7)11 (22.0)12 (24.0)23 (23.0)VI3 (5.4)6 (13.6)7 (14.0)2 (4.0)9 (9.0)VII2 (3.6)0 (0)1 (2.0)1 (2.0)2 (2.0)Total56 (100)44 (100)50 (100)50 (100)100 (100)Number (%).

### 3.4. Surface Area of Site of Tibialis Posterior Tendon Attachment

Surface areas were 120 ± 69.9 mm^2^ (25.7%) for NB, 182.4 ± 50.0 mm^2^ (39.0%) for MCB, 48.6 ± 24.7 mm^2^ (10.4%) for LCB, and the three main attachment sites provided a mean of 75.1% of the total attachment (Table 4). The ratio of the three main attachment sites (NB, MCB, and LCB) in each type was about 90% in Types II and III, and 70–80% in Types IV–VII (Table 5).

## 4. Discussion

To the best of our knowledge, this study was the first not only to examine the attachment site but also to quantify the effect of TPT on each attachment site by examining the surface area of the attachment region.

In this study, TPT was classified into seven types and four subtypes (A–D) according to the number of attachment sites. In addition, no significant difference by sex or laterality was seen between types. Previous studies have reported different results from this study [4,5,6,7,8]. In particular, previous studies using a similar classification method were 80 feet (European populations) [7] and 118 feet (Korean populations) [8], but the results were also different from this study. Therefore, it was suggested that there is high morphological variability of the TPT in relation to the insertion location, along with the possibility of significant differences according to race and gender.

In this study, attachment to the NB, MCB, and LCB, collectively representing the main site of attachment, was observed in all specimens (100%). Furthermore, the surface area of these main attachment sites (NB, MCB, and LCB) accounted for 75.1% of the total, and even among the different types, consistently accounted for more than 70% of the total regardless of the number of additional attachment sites. According to previous studies [4,5,6,7], the main attachment sites of the TPT were still the NB, MCB, and LCB, but only the proportion of attachment sites was examined, and how much of the TPT was attached to each site was not examined. The quantitative results obtained in the present study confirmed the NB, MCB, and LCB as the main sites of TPT attachment, although individual differences in attachment sites exist, further developing the findings of previous studies.

The TPT elevates the medial arch and inverts, adducts, and plantar-flexes the foot [10]. TPTD is the prevailing cause of AAFD, which is characterized by a collapse of the medial longitudinal arch [2,11]. Swanton et al. [12] reported an extension onto the MCB from the anterior band, naming this as the “navicular cuneiform ligament”. This forms a static restraint between two bony insertions (NB and MCB) and increases the lever arm of the TPT. Gwani et al. [13] also clarified three relationships between the medial and lateral longitudinal arches and the lateral arch. Deformation of the medial longitudinal arch reportedly affects other arches. In addition, the lateral arch has been reported to comprise three cuneiform bones and the CB [14]. From those previous studies and the results of this study, attachment to the NB and MCB was considered to be related to the function of the medial longitudinal arch, while attachment to the LCB appears to be related to the function of the lateral arch. Furthermore, although the relationship with muscle was not examined in this study, fibrotic binding was observed between TP and PLT (about 30%) [15], suggesting stabilization of the Lisfranc joint and involvement of the lateral arch.

Traditionally, TPTD has been understood as the main cause of adult acquired flatfoot deformity (AAFD) [2,16]. Therefore, surgical treatment has prioritized procedures that augment or replace TPT, and flexor digitorum longus (FDL) tendon transfer to the navicular bone has been most used. However, recent studies have raised considerable uncertainty regarding the ability to correct deformities using FDL tendon transfer [17]. Recently, satisfactory outcomes have been reported by transferring the flexor hallucis longus (FHL) tendon to the base of the first metatarsal bone, which is the distal portion of the medial cuneiform to secure the stability of the entire medial longitudinal arch of the foot [18]. In this study, attachment to the NB, MCB, and LCB, collectively representing the main site of attachment, was observed in all specimens (100%). Furthermore, the surface area of these main attachment sites (NB, MCB, and LCB) accounted for 75.1% of the total, and even among the different types, consistently accounted for more than 70% of the total regardless of the number of additional attachment sites. Therefore, the anatomical classification and morphological characteristics of TPT derived from this study may provide surgeons with basic knowledge and rationale for establishing the concept of stabilizing not only the medial longitudinal arch, but also the lateral longitudinal arch in tendon transfer procedures for surgical treatment of AAFD.

Some limitations need to be considered when interpreting the findings from this study. First, since only Japanese cadavers were used, potential differences between different ethnicities were not examined. The existence of ethnic differences in foot muscles has been suggested in several papers [4,19]. Caucasian individuals were found to be nearly three times more likely to show tendinopathic findings when compared to African American individuals according to a study using ultrasound [20]. Additionally, a previous study suggested high morphological variability of the TPT in relation to the insertion location, along with the possibility of significant differences according to ethnic group and gender [8]. Future studies will therefore need to consider comparisons between different ethnicities to clarify the potential for variations in TPT attachment sites. Second, in type classifications for the TPT, only attachments to bone were considered. Previous studies have reported attachments to the abductor hallucis, flexor hallucis brevis, peroneus longus tendon, spring ligament, and plantar calcaneocuboid ligament [5,6,7]. Interestingly, a finding of the PLT reserved a slip from the posterior tibialis tendon of about 30% was also reported [15]. Type classification thus needs to be performed with consideration of not only attachment to bone, but also attachment to muscles and ligaments. Third, this study evaluated the main types of insertions without indicating the presence of accessory bands.

## 5. Conclusions

The results of this study confirmed TPT attachments to the NB, MCB, and LCB in all specimens, and the surface area of these attachment sites occupied 75.1% of the total attachments to bone. Regardless of Type classification for the site of TPT attachment (number of adhered sites), attachment of the TPT to the NB, MCB, and LCB may thus provide the primary contribution to the stability of the foot arch. In the future, comparisons between ethnic groups and classification in consideration of attachments to muscles and ligaments will be needed in addition to attachments to bone.

## Figures and Tables

**Figure 1 ijerph-19-16510-f001:**
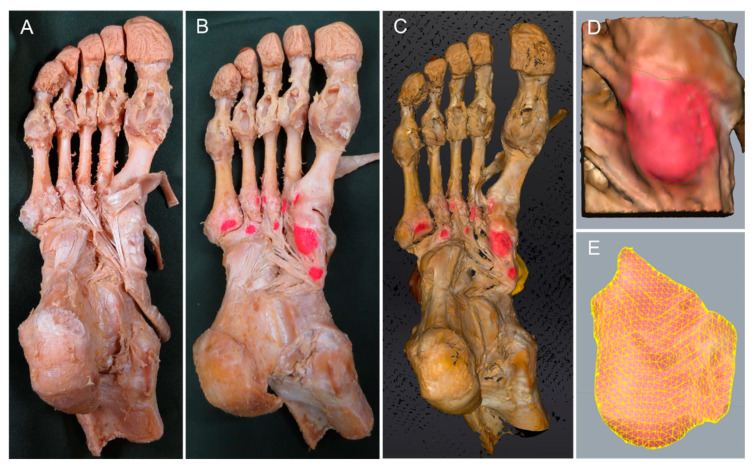
Method for measuring site of attachment. (**A**) Site of attachment of the tibialis posterior tendon: right foot, plantar view. (**B**) The site of attachment was identified by peeling the adherent tissue, then coloring the site with red pencil. (**C**) The surface area was measured using a 3D scanner to make the foot sample three-dimensional. (**D**) Enlarged view of the medial cuneiform bone. A curve is drawn as the boundary of the site of attachment of the navicular bone with a pen-type device. (**E**) Surface area was calculated using Rhinoceros 3D software.

**Figure 2 ijerph-19-16510-f002:**
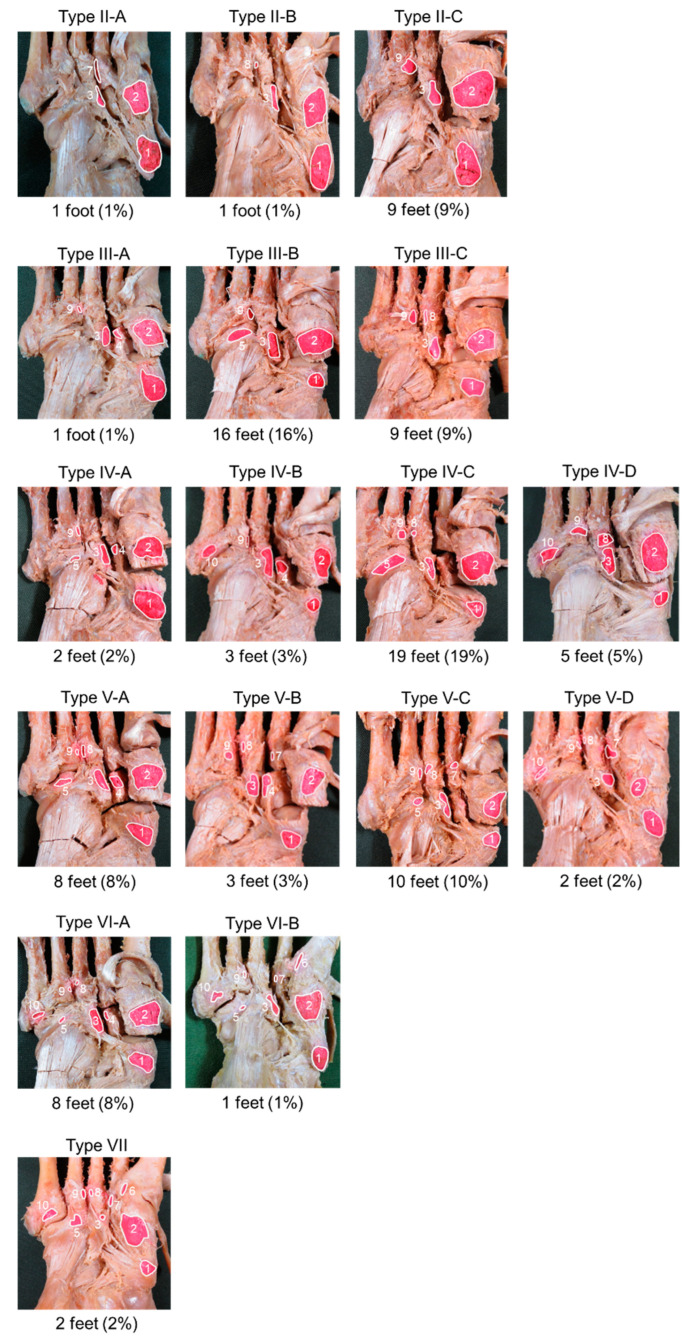
Type classification of the tibialis posterior tendon. 1, Navicular bone; 2, medial cuneiform bone; 3, lateral cuneiform bone; 4, intermediate cuneiform bone; 5, cuboid bone; 6, 1st metatarsal bone; 7, 2nd metatarsal bone; 8, 3rd metatarsal bone; 9, 4th metatarsal bone; 10, 5th metatarsal bone; M, medial; L, lateral. Type II: In addition to Type I (navicular bone, medial cuneiform bone, and lateral cuneiform bone), there is one additional site of attachment, with attachment to a total of four sites. A: 2nd metatarsal bone. B: 3rd metatarsal bone. C: 4th metatarsal bone. Type III: In addition to Type I, there are two additional sites of attachment, with attachment to a total of five sites. A: Intermediate cuneiform bone, 4th metatarsal bone. B: Cuboid bone, 4th metatarsal bone. C: 3rd metatarsal bone, 4th metatarsal bone. Type IV: In addition to Type I, there are three additional sites of attachment, with attachment to a total of six sites. A: Intermediate cuneiform bone, cuboid bone, 4th metatarsal bone. B: Intermediate cuneiform bone, 4th metatarsal bone, 5th metatarsal bone. C: Cuboid bone, 3rd metatarsal bone, 4th metatarsal bone. D: Third metatarsal bone, 4th metatarsal bone, 5th metatarsal bone. Type V: In addition to Type I, there are four additional sites of attachment, with attachment to a total of seven sites. A: Intermediate cuneiform bone, cuboid bone, 3rd metatarsal bone, 4th metatarsal bone. B: Intermediate cuneiform bone, 2nd metatarsal bone, 3rd metatarsal bone, 4th metatarsal bone. C: Cuboid bone, 2nd metatarsal bone, 3rd metatarsal bone, 4th metatarsal bone. D: 2nd metatarsal bone, 3rd metatarsal bone, 4th metatarsal bone, 5th metatarsal bone. Type VI: In addition to Type I, there are five additional sites of attachment, with attachment to a total of eight sites. A: Intermediate cuneiform bone, cuboid bone, 3rd metatarsal bone, 4th metatarsal bone, 5th metatarsal bone. B: Cuboid bone, 1st metatarsal bone, 2nd metatarsal bone, 4th metatarsal bone, 5th metatarsal bone. Type VII: In addition to Type I, there are six additional sites of attachment, with attachment to a total of nine sites. Cuboid bone, 1st metatarsal bone, 2nd metatarsal bone, 3rd metatarsal bone, 4th metatarsal bone, 5th metatarsal bone.

**Table 1 ijerph-19-16510-t001:** Site of tibialis posterior attachment and ratio.

Site of Attachment	Ratio (%)	Site of Attachment	Ratio (%)
NB	100	1 MB	4
MCB	100	2 MB	42
ICB	25	3 MB	52
LCB	100	4 MB	95
CB	66	5 MB	15

NB, navicular bone; MCB, medial cuneiform bone; ICB, intermediate cuneiform bone; LCB, lateral cuneiform bone; CB, cuboid bone; 1–5 MB, 1st–5th metatarsal bone.

**Table 2 ijerph-19-16510-t002:** Type classification for site of tibialis posterior tendon attachment.

Type			Site of Attachment	Number of Feet	Total
I	NB, MCB, LCB			0	0
II	Type I and 1 site	A	2 MB	1	11
		B	3 MB	1	
		C	4 MB	9	
III	Type I and 2 sites	A	Two of ICB, 2 MB, 3 MB or 4 MB	1	26
		B	Two of CB, 2 MB, 3 MB or 4 MB	16	
		C	Two of 2 MB, 3 MB, 4 MB or 5 MB	9	
IV	Type I and 3 sites	A	ICB, CB and 4th MB	2	29
		B	ICB, two of 2 MB, 3 MB, 4 MB or 5 MB	3	
		C	CB, two of 2 MB, 3 MB, 4 MB or 5 MB	19	
		D	Three of 2 MB, 3 MB, 4 MB or 5 MB	5	
V	Type I and 4 sites	A	ICB, CB and two of 2 MB, 3 MB, 4 MB or 5 MB	8	23
		B	ICB, 2 MB, 3 MB and 4 MB	3	
		C	CB, three of 1 MB, 2 MB, 3 MB, 4 MB or 5 MB	10	
		D	2 MB, 3 MB, 4 MB and 5 MB	2	
VI	Type I and 5 sites	A	ICB, CB, 3 MB, 4 MB and 5 MB	8	9
		B	CB, four of 1 MB, 2 MB, 3 MB, 4 MB or 5 MB	1	
VII	Type I and 6 sites		CB, 1 MB, 2 MB, 3 MB, 4 MB and 5 MB	2	2

NB, navicular bone; MCB, medial cuneiform bone; ICB, intermediate cuneiform bone; LCB, lateral cuneiform bone; CB, cuboid bone; 1 MB, 1st metatarsal bone; 2 MB, 2nd metatarsal bone; 3 MB, 3rd metatarsal bone; 4 MB, 4th metatarsal bone; 5 MB, 5th metatarsal bone.

**Table 4 ijerph-19-16510-t004:** Surface areas at sites of tibialis posterior tendon attachment.

Attachment Site	NB	MCB	ICB	LCB	CB	1 MB	2 MB	3 MB	4 MB	5 MB
Surface area (mm^2^)	120.0 ± 69.9	182.4 ± 50.0	18.4 ± 18.3	48.6 ± 24.7	17.9 ± 15.0	12.1 ± 2.9	19.5 ± 13.8	11.7 ± 9.9	16.6 ± 10.2	20.4 ± 11.6
Ratio (%)	25.7	39.0	3.9	10.4	3.8	2.6	4.2	2.5	3.6	4.4

NB, navicular bone; MCB, medial cuneiform bone; ICB, intermediate cuneiform bone; LCB, lateral cuneiform bone; CB, cuboid bone; 1 MB, 1st metatarsal bone; 2 MB, 2nd metatarsal bone; 3 MB, 3rd metatarsal bone; 4 MB, 4th metatarsal bone; 5 MB, 5th metatarsal bone.

**Table 5 ijerph-19-16510-t005:** Percentage surface areas for each type at navicular bone, medial cuneiform bone, and lateral cuneiform bone.

Type	NB	MCB	LCB	Total
II	40.5	43.1	8.7	92.3
III	32.6	38.3	9.8	80.6
IV	22.7	42.6	11.5	76.8
V	27.1	34.1	10.7	71.9
VI	20.5	42.9	14.5	78.0
VII	9.9	60.9	0.7	71.5

NB, navicular bone; MCB, medial cuneiform bone; LCB, lateral cuneiform bone.

## Data Availability

The datasets used and/or analyzed during the current study are available from the corresponding author on reasonable request.

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
