# Peer review of "Anatomical Study of Sites and Surface Area of the Attachment Region of Tibial Posterior Tendon Attachment"

_ijerph, 2022, doi:10.3390/ijerph192416510_

Round 1

Reviewer 1 Report

I reviewed the article entitled: “Anatomical study of sites and footprints of tibial posterior tendon attachment”. This study examines the attachment site and also to quantify the effect of the tibialis posterior tendon on each attachment site of the foot by examining the surface area of the attachment region.

The manuscript presents an anatomical descriptive analysis, however, contains some important deficiencies that, due to this, make me not recommend the manuscript for publication in the International Journal of Environmental Research and Public Health.

Below are some appointments of the article:

*Lines 66-67 and 194-195. “The purpose of this study was to use the largest sample size compared to previous studies […]”. This study analyzed 100 feet from 50 cadavers, and as the authors discuss in this manuscript, Musial and Prackka (lines 39-42) examined 122 feet. So the authors’ statement is not correct.

*It is not clear what is the aim of this study (it is only a descriptive study). The Authors base their arguments saying that the sample size of previous studies was small.

*This study only presents results of descriptive statistics (frequency and percentage). Despite the fact that in material and methods the authors state that they performed the Chi-square statistic, the data from this test is not presented in the text and/or tables.

*Line 122. Do not add the legend of Table 1 in the subheading.

Author Response

December 2, 2022

Editorial Board

Ref: Submission ID ijerph-2067416
“Anatomical study of sites and sites and surface area of tibial posterior tendon attachment” by Mutsuaki Edama

Dear Reviewer 1:

Thank you for your letter. We are grateful for the detailed feedback provided by the reviewers, which we feel has helped us to significantly improve the paper. Attached are our point-by-point responses to the reviewers’ comments and our revised manuscript, which we hope will now meet with your approval. For your convenience, we have attached a copy of the manuscript with all revisions highlighted in red font. We believe that our revisions have addressed the issues raised by the reviewers and trust that the manuscript is now suitable for publication in International Journal of Environmental Research and Public Health.

Thank you again for your thoughtful comments, and we look forward to hearing from you soon.

Sincerely,

Mutsuaki Edama

RESPONSE TO REVIEWER #1

  1. Lines 66-67 and 194-195. “The purpose of this study was to use the largest sample size compared to previous studies […]”. This study analyzed 100 feet from 50 cadavers, and as the authors discuss in this manuscript, Musial and Prackka (lines 39-42) examined 122 feet. So the authors’ statement is not correct.

→The text has been corrected as suggested. (Lns 12-14; Lns 71-73; Lns 240-242)

・Abstract Lns 12-14

Background: The purpose of this study was not only to examine the attachment site but also to quantify the effect of the tibialis posterior tendon (TPT) on each attachment site by examining the surface area of the attachment region.

・Introduction Lns 71-73

The purpose of this study was not only to examine the attachment site but also to quantify the effect of TPT on each attachment site by examining the surface area of the attachment region.

・Discussion Lns 240-242

To the best of our knowledge, this study was the first not only to examine the attachment site but also to quantify the effect of TPT on each attachment site by examining the surface area of the attachment region.

  1. It is not clear what is the aim of this study (it is only a descriptive study). The Authors base their arguments saying that the sample size of previous studies was small.

→The text has been corrected as suggested. (Lns 71-73)

・Introduction Lns 71-73

The purpose of this study was not only to examine the attachment site but also to quantify the effect of TPT on each attachment site by examining the surface area of the attachment region.

  1. This study only presents results of descriptive statistics (frequency and percentage). Despite the fact that in material and methods the authors state that they performed the Chi-square statistic, the data from this test is not presented in the text and/or tables.

→The text has been corrected as suggested. (Lns 215-216)

Results Lns 215-216

Differences in laterality and sex by type

No significant differences were observed between types (Table 3).

Table 3. Differences in laterality and sex for each type

Type

Male

Female

Right

Left

Total

I

0 (0)

0 (0)

0 (0)

0 (0)

0 (0)

II

6 (10.7)

5 (11.4)

6 (12.0)

5 (10.0)

6 (10.7)

III

15 (26.8)

11 (25.0)

13 (26.0)

13 (26.0)

15 (26.8)

IV

17 (30.4)

12 (27.3)

12 (24.0)

17 (34.0)

17 (30.4)

V

13 (23.2)

10 (22.7)

11 (22.0)

12 (24.0)

13 (23.2)

VI

3 (5.4)

6 (13.6)

7 (14.0)

2 (4.0)

3 (5.4)

VII

2 (3.6)

0 (0)

1 (2.0)

1 (2.0)

2 (3.6)

Total

56 (100)

44 (100)

50 (100)

50 (100)

56 (100)

Number (%)

  1. Line 122. Do not add the legend of Table 1 in the subheading.

→The text has been corrected as suggested. (Table 1)

Table 1. Site of tibialis posterior attachment and ratio

Site of attachment

Ratio (%)

Site of attachment

Ratio (%)

NB

100

1MB

4

MCB

100

2MB

42

ICB

25

3MB

52

LCB

100

4MB

95

CB

66

5MB

15

NB, navicular bone; MCB, medial cuneiform bone; ICB, intermediate cuneiform bone; LCB, lateral cuneiform bone; CB, cuboid bone; 1-5MB, 1st–5th metatarsal bone.

Reviewer 2 Report

The authors provide valuable information on the variability of the attachment of the posterior tibial tendon that is worth publishing. A uniform classification in this case is necessary and could be helpful in medical-surgical practice. Despite the positive aspects of this anatomical study, the manuscript needs to be thoroughly revised before publication. All recommendations are listed below. 

- First, I would like to ask the authors to reconsider the use of the term "footprint" as it refers to the imprint left by a foot, its plantar size weighed by the body, which could be confusing especially in this foot-related terminology. I suggest using impression, imprint instead.

- What do the authors mean by the "largest sample size" given in the abstract? Because as mentioned earlier, the sample consisted of 50 cadavers, which is not a large sample. As mentioned, it is larger than some previous studies, but for example, the Olewnik 2019 study used 80 cadavers. Please revise the study carefully.

- In the Material and Method section - was surgery the only exclusion criterion? Were no other factors considered? Could these other factors significantly affect the TPT attachments? Please explain.

- Does your study evaluate the main types of insertions without indicating the presence of accessory bands? If so, this should be clearly stated in the objective of the study.

- It seems that the authors want to conclude that sample size has a stronger influence on attachment variability than gender or laterality. This claim is too presumptuous and should be stated accordingly when examining other factors - gender, laterality, ethnicity, perhaps other foot or muscle deformities. Please emphasize this in your paper.

- How might your results affect foot type and arch, in terms of rigidity and subtility of the medial and lateral arch, as you state in your conclusion. Does this apply to both arches of the foot or just the medial arch?

- Could the results of your study be helpful in surgical practice when it comes to flexor digitorum longus transfer? Please mention this briefly in the Discussion and Conclusion section.

- In the conclusion section, I would avoid the term "race" and use "ethnicity" or "ethnic group" instead.

Author Response

December 2, 2022

Editorial Board

Ref: Submission ID ijerph-2067416
“Anatomical study of sites and sites and surface area of tibial posterior tendon attachment” by Mutsuaki Edama

Dear Reviewer 2:

Thank you for your letter. We are grateful for the detailed feedback provided by the reviewers, which we feel has helped us to significantly improve the paper. Attached are our point-by-point responses to the reviewers’ comments and our revised manuscript, which we hope will now meet with your approval. For your convenience, we have attached a copy of the manuscript with all revisions highlighted in red font. We believe that our revisions have addressed the issues raised by the reviewers and trust that the manuscript is now suitable for publication in International Journal of Environmental Research and Public Health.

Thank you again for your thoughtful comments, and we look forward to hearing from you soon.

Sincerely,

Mutsuaki Edama

RESPONSE TO REVIEWER #2

  1. First, I would like to ask the authors to reconsider the use of the term "footprint" as it refers to the imprint left by a foot, its plantar size weighed by the body, which could be confusing especially in this foot-related terminology. I suggest using impression, imprint instead.

→The text has been used the surface area of the attachment region instead of footprint as suggested. (Title; Key words)

Title

Anatomical study of sites and surface area of the attachment region of tibial posterior tendon attachment

Key words

Key words: tibialis posterior tendon dysfunction; attachment site; surface area region

  1. What do the authors mean by the "largest sample size" given in the abstract? Because as mentioned earlier, the sample consisted of 50 cadavers, which is not a large sample. As mentioned, it is larger than some previous studies, but for example, the Olewnik 2019 study used 80 cadavers. Please revise the study carefully.

→The text has been corrected as suggested. (Lns 12-14; Lns 71-73; Lns 240-242)

・Abstract Lns 12-14

Background: The purpose of this study was not only to examine the attachment site but also to quantify the effect of the tibialis posterior tendon (TPT) on each attachment site by examining the surface area of the attachment region.

・Introduction Lns 71-73

The purpose of this study was not only to examine the attachment site but also to quantify the effect of TPT on each attachment site by examining the surface area of the attachment region.

・Discussion Lns 240-242

To the best of our knowledge, this study was the first not only to examine the attachment site but also to quantify the effect of TPT on each attachment site by examining the surface area of the attachment region.

  1. In the Material and Method section - was surgery the only exclusion criterion? Were no other factors considered? Could these other factors significantly affect the TPT attachments? Please explain.

→As stated,“a sign of previous major surgery around the ankle”was used as an exclusion criterion.

  1. Does your study evaluate the main types of insertions without indicating the presence of accessory bands? If so, this should be clearly stated in the objective of the study.

→The limitation has been corrected as suggested. (Lns 292-308)

Some limitations need to be considered when interpreting the findings from this study. First, since only Japanese cadavers were used, potential differences between differ-ent ethnicities were not examined. The existence of ethnic differences in foot muscles has been suggested in several papers [4,19]. Caucasian individuals were found to be nearly three times more likely to show tendinopathic findings when compared to Afri-can-American individuals according to a study using ultrasound [20]. And previous study suggested high morphological variability of the TPT in relation to the insertion loca-tion, along with the possibility of significant differences according to ethnic group and gender [8]. Future studies will therefore need to consider comparisons between different ethnicities to clarify the potential for variations in TPT attachment sites. Second, in type classifications for the TPT, only attachments to bone were considered. Previous studies have reported attachments to the abductor hallucis, flexor hallucis brevis, peroneus lon-gus tendon, spring ligament, and plantar calcaneocuboid ligament [5-7]. Interestingly, a finding of the PLT reserved a slip from the posterior tibialis tendon about 30% was also reported [15]. Type classification thus needs to be performed with consideration of not only attachment to bone, but also attachment to muscles and ligaments. Third, this study evaluated the main types of insertions without indicating the presence of accessory bands.

[4] Willegger, M.; Seyidova, N.; Schuh, R.; Windhager, R.; Hirtler, L. The tibialis posterior tendon footprint: an anatomical dissection study. Journal of foot and ankle research 2020, 13, 25.

[5] Bloome, D.M.; Marymont, J.V.; Varner, K.E. Variations on the insertion of the posterior tibialis tendon: a cadaveric study. Foot & ankle international. / American Orthopaedic Foot and Ankle Society [and] Swiss Foot and Ankle Society 2003, 24, 780-783.

[6] Musial, W.; Pracka, H. [VALUE OF VECTORCARDIOGRAPHY IN THE DIAGNOSIS OF VENTRICULAR HYPERTROPHY IN ACQUIRED CARDIAC DEFECTS]. Kardiol Pol 1963, 46, 97-103.

[7] Olewnik, Ł. A proposal for a new classification for the tendon of insertion of tibialis posterior. Clin Anat 2019, 32, 557-565.

[8] Park, J.H.; Kim, D.; Kwon, H.W.; Lee, M.; Choi, Y.J.; Park, K.R.; Youn, K.H.; Cho, J. A New Anatomical Classification for Tibialis Posterior Tendon Insertion and Its Clinical Implications: A Cadaveric Study. Diagnostics (Basel) 2021, 11.

[15] Edama, M.; Takabayashi, T.; Hirabayashi, R.; Yokota, H.; Inai, T.; Sekine, C.; Matsuzawa, K.; Otsuki, T.; Maruyama, S.; Kageyama, I. Anatomical variations in the insertion of the peroneus longus tendon. Surgical and radiologic anatomy: SRA 2020, 42, 1141-1144.

[19] Edama, M.; Kubo, M.; Onishi, H.; Takabayashi, T.; Yokoyama, E.; Inai, T.;

Watanabe, H.; Nashimoto, S.; Kageyama, I. Anatomical study of toe flexion by

flexor hallucis longus. Annals of anatomy = Anatomischer Anzeiger: official organ

of the Anatomische Gesellschaft 2016, 204, 80-85.

[20] Mills, F.B.t.; Williams, K.; Chu, C.H.; Bornemann, P.; Jackson, J.B., 3rd. Prevalence of Abnormal Ultrasound Findings in Asymptomatic Posterior Tibial Tendons. Foot & ankle international. / American Orthopaedic Foot and Ankle Society [and] Swiss Foot and Ankle Society 2020, 41, 1049-1055.

  1. It seems that the authors want to conclude that sample size has a stronger influence on attachment variability than gender or laterality. This claim is too presumptuous and should be stated accordingly when examining other factors - gender, laterality, ethnicity, perhaps other foot or muscle deformities. Please emphasize this in your paper.

→The text has been corrected as suggested. (Lns 243-250)

Discussion

In this study, TPT was classified into seven types and four subtypes (A–D) according to the number of attachment sites. In addition, no significant difference by sex or laterality was seen between types. Previous studies have reported different results from this study [4-8]. In particular, previous studies using a similar classification method were 80 feet (European populations) [7] and 118 feet (Korean populations) [8], but the results were also different from this study. Therefore, it was suggested that high morphological variability of the TPT in relation to the insertion location, along with the possibility of significant dif-ferences according to race and gender.

[4] Willegger, M.; Seyidova, N.; Schuh, R.; Windhager, R.; Hirtler, L. The tibialis posterior tendon footprint: an anatomical dissection study. Journal of foot and ankle research 2020, 13, 25.

[5] Bloome, D.M.; Marymont, J.V.; Varner, K.E. Variations on the insertion of the posterior tibialis tendon: a cadaveric study. Foot & ankle international. / American Orthopaedic Foot and Ankle Society [and] Swiss Foot and Ankle Society 2003, 24, 780-783.

[6] Musial, W.; Pracka, H. [VALUE OF VECTORCARDIOGRAPHY IN THE DIAGNOSIS OF VENTRICULAR HYPERTROPHY IN ACQUIRED CARDIAC DEFECTS]. Kardiol Pol 1963, 46, 97-103.

[7] Olewnik, Ł. A proposal for a new classification for the tendon of insertion of tibialis posterior. Clin Anat 2019, 32, 557-565.

[8] Park, J.H.; Kim, D.; Kwon, H.W.; Lee, M.; Choi, Y.J.; Park, K.R.; Youn, K.H.; Cho, J. A New Anatomical Classification for Tibialis Posterior Tendon Insertion and Its Clinical Implications: A Cadaveric Study. Diagnostics (Basel) 2021, 11.

  1. How might your results affect foot type and arch, in terms of rigidity and subtility of the medial and lateral arch, as you state in your conclusion. Does this apply to both arches of the foot or just the medial arch?

→From previous studies and the results of this study, attachment to the NB and MCB was considered related to the function of the medial longitudinal arch, while attachment to the LCB appears related to the function of the lateral arch. Furthermore, although the relationship with muscle was not examined in this study, fibrotic binding was observed between TP and PLT (about 30%) [15], suggesting stabilization of the Lisfranc joint and involvement of the lateral arch.

[15] Edama, M.; Takabayashi, T.; Hirabayashi, R.; Yokota, H.; Inai, T.; Sekine, C.; Matsuzawa, K.; Otsuki, T.; Maruyama, S.; Kageyama, I. Anatomical variations in the insertion of the peroneus longus tendon. Surgical and radiologic anatomy: SRA 2020, 42, 1141-1144.

  1. Could the results of your study be helpful in surgical practice when it comes to flexor digitorum longus transfer? Please mention this briefly in the Discussion and Conclusion section.

→The text has been corrected as suggested. (Lns 275-291)

Traditionally, TPTD has been understood as the main cause of adult acquired flatfoot deformity (AAFD) [2,16]. Therefore, surgical treatment has prioritized procedures that augment or replace TPT, and flexor digitorum longus (FDL) tendon transfer to the navicu-lar bone has been most used. However, recent studies have raised considerable uncertain-ty regarding the ability to correct deformities using FDL tendon transfer [17]. Recently, satisfactory outcomes have been reported by transferring the flexor hallucis longus (FHL) tendon to the base of the first metatarsal bone, which is the distal portion of the medial cuneiform to secure the stability of the entire medial longitudinal arch of the foot [18]. In this study, attachment to the NB, MCB, and LCB, collectively representing the main site of attachment, was observed in all specimens (100%). Furthermore, the surface area of these main attachment sites (NB, MCB and LCB) accounted for 75.1% of the total, and even among the different types, consistently accounted for more than 70% of the total regardless of the number of additional attachment sites. Therefore, the anatomical classification and morphological characteristics of TPT derived from this study may provide surgeons with basic knowledge and rationale for establishing the concept of stabilizing not only the me-dial longitudinal arch, but also the lateral longitudinal arch in tendon transfer procedures for surgical treatment of AAFD.

[2] Kohls-Gatzoulis, J.; Woods, B.; Angel, J.C.; Singh, D. The prevalence of symptomatic posterior tibialis tendon dysfunction in women over the age of 40 in England. Foot and ankle surgery: official journal of the European Society of Foot and Ankle Surgeons 2009, 15, 75-81.

[16] Ross, M.H.; Smith, M.; Plinsinga, M.L.; Vicenzino, B. Self-reported social and activity restrictions accompany local im-pairments in posterior tibial tendon dysfunction: a systematic review. Journal of foot and ankle research 2018, 11, 49.

[17] Vaudreuil, N.J.; Ledoux, W.R.; Roush, G.C.; Whittaker, E.C.; Sangeorzan, B.J. Comparison of transfer sites for flexor digitorum longus in a cadaveric adult acquired flatfoot model. Journal of orthopaedic research: official publication of the Orthopaedic Research Society 2014, 32.

[18] Kim, J.; Kim, J.B.; Lee, W.C. Dynamic medial column stabilization using flexor hallucis longus tendon transfer in the surgical reconstruction of flatfoot deformity in adults. Foot and ankle surgery: official journal of the European Society of Foot and Ankle Surgeons 2021, 27, 920-927.

  1. In the conclusion section, I would avoid the term "race" and use "ethnicity" or "ethnic group" instead.

→The text has been corrected as suggested. (conclusion)

CONCLUSION

The results of this study confirmed TPT attachments to the NB, MCB, and LCB in all specimens, and the surface area of these attachment sites occupied 75.1% of the total attachments to bone. Regardless of Type classification for site of TPT attachment (number of adhered sites), attachment of the TPT to the NB, MCB, and LCB may thus provide the primary contribution to the stability of the foot arch. In the future, comparisons between ethnic group and classification in consideration of attachments to muscles and ligaments will be needed in addition to attachments to bone.

Round 2

Reviewer 1 Report

The revised version of manuscript is improved; the Authors incorporated the suggestions provided by the reviewers. I thank the Authors for the great effort they have made in responding to the comments of the reviewers. The manuscript can be accepted for publication into “International Journal of Environmental Research and Public Health” after minor revision.

Despite the fact that in material and methods the authors state that they performed the Chi-square statistic, the data from this test is not presented in the revised manuscript. Authors added the following sentence (line 216): “No significant differences were observed between types (Table 3).” However, the Chi-square values, degrees of freedom and p-values were not provided. These values MUST be provided in Table 3 or main text.

In addition, some errors must be corrected prior to accept the manuscript for publication. In Table 3, the values in the “Total” column are the same values than “Male” column.

Author Response

RESPONSE TO REVIEWER #1

  1. Despite the fact that in material and methods the authors state that they performed the Chi-square statistic, the data from this test is not presented in the revised manuscript. Authors added the following sentence (line 216): “No significant differences were observed between types (Table 3).” However, the Chi-square values, degrees of freedom and p-values were not provided. These values MUST be provided in Table 3 or main text.

→The text has been corrected as suggested. (Lns 215-217)

3.3. Differences in sex and laterality by type (Table 3)

 There were no significant differences between male and female (p=0.613) and between the right and left legs (p=0.582) for each Types.

  1. In addition, some errors must be corrected prior to accept the manuscript for publication. In Table 3, the values in the “Total” column are the same values than “Male” column.

→The text has been corrected as suggested. (Table 3)

Table 3. Differences in laterality and sex for each type

Type

Male

Female

Right

Left

Total

I

0 (0)

0 (0)

0 (0)

0 (0)

0 (0)

II

6 (10.7)

5 (11.4)

6 (12.0)

5 (10.0)

11 (11.0)

III

15 (26.8)

11 (25.0)

13 (26.0)

13 (26.0)

26 (26.0)

IV

17 (30.4)

12 (27.3)

12 (24.0)

17 (34.0)

29 (29.0)

V

13 (23.2)

10 (22.7)

11 (22.0)

12 (24.0)

23 (23.0)

VI

3 (5.4)

6 (13.6)

7 (14.0)

2 (4.0)

9 (9.0)

VII

2 (3.6)

0 (0)

1 (2.0)

1 (2.0)

2 (2.0)

Total

56 (100)

44 (100)

50 (100)

50 (100)

100 (100)

Number (%)
